# Lyotropic Liquid Crystalline Nanostructures as Drug Delivery Systems and Vaccine Platforms

**DOI:** 10.3390/ph15040429

**Published:** 2022-03-31

**Authors:** Maria Chountoulesi, Stergios Pispas, Ioulia K. Tseti, Costas Demetzos

**Affiliations:** 1Section of Pharmaceutical Technology, Department of Pharmacy, School of Health Sciences, National and Kapodistrian University of Athens, Panepistimioupolis Zografou, 15771 Athens, Greece; mchountoules@pharm.uoa.gr; 2Theoretical and Physical Chemistry Institute, National Hellenic Research Foundation, 48 Vassileos Constantinou Avenue, 11635 Athens, Greece; pispas@eie.gr; 3Uni-Pharma S.A., 14564 Kifissia, Greece; jtsetis@uni-pharma.gr

**Keywords:** lyotropic liquid crystals, cubosomes, vaccines, drug delivery nanosystems, stimuli-responsive, lipid nanoparticles, controlled drug release

## Abstract

Lyotropic liquid crystals result from the self-assembly process of amphiphilic molecules, such as lipids, into water, being organized in different mesophases. The non-lamellar formed mesophases, such as bicontinuous cubic (cubosomes) and inverse hexagonal (hexosomes), attract great scientific interest in the field of pharmaceutical nanotechnology. In the present review, an overview of the engineering and characterization of non-lamellar lyotropic liquid crystalline nanosystems (LLCN) is provided, focusing on their advantages as drug delivery nanocarriers and innovative vaccine platforms. It is described that non-lamellar LLCN can be utilized as drug delivery nanosystems, as well as for protein, peptide, and nucleic acid delivery. They exhibit major advantages, including stimuli-responsive properties for the “on demand” drug release delivery and the ability for controlled release by manipulating their internal conformation properties and their administration by different routes. Moreover, non-lamellar LLCN exhibit unique adjuvant properties to activate the immune system, being ideal for the development of novel vaccines. This review outlines the recent advances in lipid-based liquid crystalline technology and highlights the unique features of such systems, with a hopeful scope to contribute to the rational design of future nanosystems.

## 1. Introduction

A plethora of research has been carried out on lyotropic liquid crystalline nanostructures and their advantageous applications as drug delivery vehicles. Among the different lyotropic liquid crystalline mesophases, the non-lamellar ones, consisting of well-defined networks of aqueous channels and lipid bilayer membranes, namely, the bicontinuous cubic and inverse hexagonal mesophases, exhibit significant scientific research and experience rapid technological developments in the last decades. They can be applied either as bulk phases or as fabricated colloidal nanocarriers, e.g., cubosomes and hexosomes.

Non-lamellar lyotropic liquid crystalline nanosystems (LLCN) can be utilized as drug, protein, peptide, or nucleic acid delivery nanosystems, being able to host a wide variety of hydrophilic, hydrophobic, and amphiphilic small molecules and biomacromolecules. They exhibit major advantages, including the ability of controlled release, providing that that a careful design has been carried out to manipulate the formation of their internal organization. They can be administrated by various routes, resulting in enhanced drug bioavailability and therapeutic efficiency. Moreover, they can be easily functionalized by the incorporation of smart biomaterials, e.g., stimuli responsive molecules, surfactants, or polymers, towards the development of “on-demand” drug release delivery vehicles.

In the present review, an overview of the engineering and characterization of non-lamellar LLCN is provided, focusing on their advantages as drug delivery nanocarriers for controlled release. Different case studies are examined that describe the delivery of various therapeutic agents, being administrated by different routes, as well as examples of vaccines. The design of stimuli-responsive non-lamellar LLCN is also described. The impact of the formulation parameters, the interactions taking place among the biomaterials, and the environmental parameters on the formation/arrangement of their internal nanostructure and thereby on the resulting behavior as drug carriers is also highlighted. Furthermore, this review outlines the recent advances in the liquid crystalline technology and emphasizes on the unique features of these systems that make them promising platforms for drug delivery, describing different strategies towards the manipulation of their nanostructure for the modification of the drug release kinetics, with a further aim to contribute to the rational design of future systems in forthcoming studies.

## 2. Self-Assembly and Formation of Liquid Crystalline Nanostructures

Amphiphilic molecules, such as the lipids, being dispersed into water, have the ability for self-assembly into different mesophases, as a function of concentration and temperature, eventually forming lyotropic liquid crystalline mesophases [1,2]. The molecules position themselves in such a way in order to minimize the free energy of the system, by exposing the hydrophilic regions to the aqueous environment and tightly packing their hydrophobic domains in the interior as an effort to minimize the interface with the existing solvent. The type of the self-assembled nanostructure is mainly governed by the surfactant molecular shape that can be predicted by the critical packing parameter (CPP) (CPP = v/(A ∗ l)). The CPP is a geometrical value consisting of the ratio between the volume of the hydrophobic lipid tail (v) and the product of the cross-sectional lipid head area (A) and the lipid chain length (l). Taking into account the changes of the CPP, order–order transitions associated with the change in the curvature of the water–lipid interface can be forecasted. Cylindrical shaped surfactant molecules (CPP ≈ 1) tend to form planar membranes (fluid lamellar (L_a_) phase), and conversely, cone-shaped (CPP < 1) and wedge-shaped surfactant molecules (CPP > 1) prefer the formation of convex (type 1/normal type, oil-in-water (O/W) phases) and concave (type 2/inverted type, water-in-oil (W/O) phases) interfaces, respectively (Figure 1, left panel).

The nanostructures of inverted-type/type 2 liquid crystalline phases and micellar solutions are independent of water content under full hydration conditions and therefore are stable against water dilution and particularly attractive in the development of drug delivery systems [1,4,5]. In the present review, we are focus on the inverted-type/type 2 liquid crystalline phases. Among the different mesophases some well-defined, thermodynamically stable structures can be observed, such as lamellar (*L_a_* or *L_C_* depending on whether the alkyl tail is amorphous or has crystallized), as well as inverted liquid crystalline phases, such as hexagonal (*H_II_* or *H_2_*) and bicontinuous (*Q_II_* or *Q_2_*) cubic phases [1,2,6].

The bicontinuous cubic (*Q_II_*) phase is a very interesting, complex structure, consisting of a 3D network, separating two distinct, continuous but nonintersecting, hydrophilic sections/water channels. Three types of the bicontinuous cubic (*Q_II_*) phases with different space group symmetry have been identified in various lipid systems, named as *Im3m* (primitive type), *Pn3m* (double-diamond type), and *Ia3d* (gyroid type) assemblies. Regarding the inverse hexagonal (*H_II_*) phase, it is a 2D structure consisting of water-filled cylindrical rods (hydrophilic nanochannels), each surrounded by a lipidic bilayer and following a long-range two-dimensional order, being hexagonally close densely packed in a two-dimensional continuous hydrophobic medium [1,3,7,8,9,10,11]. Moreover, the “melted” cubic mesophase, called as sponge phase (*L_3_*), has also been identified, containing a bicontinuous bilayer, but without a long-range order structure [12]. Finally, the *Fd3m* micellar cubic phase is composed of two discrete inverse micelle populations organized into a double diamond network that is isolated from the external environment, enabling the lowest diffusion coefficient among the studied mesophases. The described self-assembled lipid nanostructures are illustrated in Figure 2.

## 3. Engineering of Non-Lamellar Lyotropic Liquid Crystalline Nanoparticles (Cubosomes and Hexosomes)

### 3.1. Materials and Preparation Methods

The aforementioned non-lamellar mesophases, formed by the self-assembly process of the surfactant-like lipids in water, can be exploited towards the development of colloidal dispersions of nanoparticles, with confined internal structures of 3D well-ordered bicontinuous cubic (*Q_II_*) and 2D columnar hexagonal (*H_II_*) phases, the cubosomes and the hexosomes, respectively. The fragmentation and the steric stabilization of the above-described non-lamellar bulk mesophases to cubosomal of hexosomal nanoparticle colloidal dispersions requires the presence of an efficient amphiphilic stabilizer, which can also prevent the formation of aggregates over time, providing physicochemical stability.

Glyceryl monooleate or monoolein (GMO or MO, respectively, 2,3-dihydroxypropyl oleate) and phytantriol (PHYT, 3,7,11,15-tetramethyl-1,2,3-hexadecanetriol) are the two most popular lipids that has been proven to form cubic and hexagonal phases under specific conditions of water concentration and temperature. Their chemical structures are illustrated in Figure 1 (right panel). GMO is a polar unsaturated monoglyceride and it is characterized as a nontoxic, biodegradable, and biocompatible material being listed in the FDA’s Inactive Ingredients Guide. It has also been called a “magic lipid”, exhibiting numerous applications in the fields of drug delivery, emulsion stabilization, and protein crystallization and resulting in an extraordinary increase in publications, where it has been referred [14,15]. However, hydrolysis of the ester linkage within the monoolein headgroup, or oxidation of unsaturated bonds in the hydrocarbon backbone are potential disadvantages, yielding a restriction of long-term storage and stability. On the other hand, PHYT is a well-known, commercially available, active ingredient used in cosmetic industry for skincare products. One of the referred advantages of PHYT is its improved chemical stability compared to other fatty acid-based materials, such as GMO, in aqueous and model gastrointestinal conditions, due to the presence of the phytanyl backbone without ester and unsaturated bonds, protecting PHYT from hydrolysis and enzymatic degradation. However, GMO is generally preferred over PHYT due to a lower propensity to cause haemolysis at dose-relevant concentrations.

Other lipids known to form non-lamellar liquid crystalline mesophases are monolinolein, monoelaidin, monoelaidin, phosphatidylethanolamine, oleoylethanolamide, phospholipids, PEGylated phospholipids, alkyl glycerates, and glycolipids. Oleyl glycerate (OG, 2,3-dihydroxypropionic acid octadec-9-enyl ester) and phytanyl glycerate (PG, 2,3-dihydroxypropionic acid 3,7,11,15-tetramethyl-hexadecyl ester) are found to form hexagonal phase at physiological temperature in excess water [1,15,16].

Amphiphilic block copolymers are known to act as ideal stabilizers. One of the most efficient and well-investigated polymeric stabilizers is the amphiphilic triblock copolymer Poloxamer 407 (P407) (PEO_99_–PPO_67_–PEO_99_, where PEO: poly[ethylene oxide] and PPO: poly[propylene oxide]), which is commercially known as Pluronic^®^ F-127 (F127) (Figure 1, right panel), as well as other Pluronic polymers, such as Pluronic^®^ F-108 (F108). According to Chong et al. [17], the minimum hydrophobic domain length of the polymeric stabilizer of the family of Poloxamers that can successfully stabilize the liquid crystalline nanoparticles is 40 PPO units and a hydrophilic domain containing at least 19 PEO units. Other stabilizers that have been reported are β-casein, polysorbates (Tween 80 and Tween 85), modified cellulose, polyethoxylated castor oil (Cremophor EL), polyethoxylated phytosterols, and polyethoxylated stearates (Myrj series), as well as lipids conjugated with polymers, such as 1,2-distearoyl-sn-glycero-3-phosphoethanolamine conjugated with poly(ethylene glycol) (DSPE-PEG) [1,2,9,15,18,19,20]. Chountoulesi et al. [21] utilized for the first time gradient copolymers of poly(2-oxazoline)s as polymeric stabilizers and compared their stabilizing behavior with block copolymers consisting of poly(ε-caprolactone). Zabara et al. [22] managed to create for the first time a stabilizer-free nanocarrier, colloidally stable upon time, for the human cathelicidin derived antimicrobial peptide LL-37, by the spontaneous integration of LL-37 into GMO-based cubosomes, which were prepared by dispersing the GMO bulk phase in water using ultrasonication.

Regarding the preparation methods of the liquid crystalline nanoparticles, the top-down (TD) and bottom-up techniques (BU) are the two commonly used preparation methods. The TD manufacturing method is the most commonly employed method and requires high energy input, provided by sonication, ultrasonication, shearing, or high-pressure homogenization, that eventually leads to fragmentation of viscous hydrated lipidic precursor mixtures and provides nanoparticles of controlled size (≈100−400 nm) with low polydispersity (≈0.1−0.4). One of the most critical disadvantages of the TD method is the possible degradation of the incorporated bioactive molecules, such as the more sensitive proteins and peptides, by the heat generated during the mechanical processing. Although it is a rapid method, it is only applicable at a small scale.

The BU method was first referred to by Spicer and Hayden [23] as the hydrotrope method, due to the utilization of an auxiliary solvent, such as ethanol. In the BU method, cubosomes are formed by crystallization from precursors using low amounts of energy, while the formation of the viscous bulk phase is avoided. The hydrotrope is used to solubilize the host lipid, creating a liquid precursor that is diluted into an aqueous medium (water) resulting in the lipid droplet nucleation and spontaneous formation of the nanoparticles. Although the BU method uses milder energy than the TD method, it results in less controllable sizes and residues of solvent in the final formulation, which should be totally removed in the case of all medical applications. However, the main advantages of this method, compared to the TD technique, is that less energy is used, and the processing is being carried out also at ambient temperatures, which allows for the large-scale preparation of cubosomes, with a small fraction of present vesicles and also the incorporation of sensitive compounds [6,9,24,25,26,27].

### 3.2. Characterization Methods

Small angle X-ray scattering (SAXS) is one of the most usually utilized techniques for characterizing the liquid crystalline phases of the nanoparticles and their corresponding internal organization, referred also as the “gold standard”. On the basis of Bragg’s law, the scattered radiation satisfying the constructive condition would lead to strong intensities in the diffraction pattern. These intensities, known as Bragg peaks, can be used to identify the structure of the liquid crystalline mesophase, because each phase has a signature of Bragg peak spacing ratios. The actual peak positions also provide information about the size of the crystal lattice. Various formulation factors including lipid or stabilizer composition and the effect of the solubilization of hydrophilic or hydrophobic drugs, as well as environmental factors such as the temperature, pressure, and environmental pH, can be investigated with the SAXS technique. Therefore, SAXS can detect rapid phase changes, curvature variations, and the structural dynamics of self-assembled lipidic systems through in situ experiments [1,2,25,26].

Cryogenic transmission electron microscopy (cryo-TEM) is considered to be the most accurate method in order to illustrate exact external and internal morphology of LLCN. As Kuntsche et al. [28] describe, the major advantage of cryo-TEM is the fact that after plunge freezing of dispersions provides the direct illustration of colloids in the vitrified, frozen-hydrated state, therefore being much closer to their native state, also revealing information about their internal and three-dimensional structure, as well as the shape (e.g., multilamellar vesicles, cubosomes, hexosomes). Liquid crystalline nanoparticles can only self-assemble in the presence of a solvent (typically water), and thus the use of cryo-TEM, where the samples are vitrified preserving their structure, is superior to conventional TEM, where water should be removed and therefore the internal structure most probably will be altered [2,28].

Moreover, cryo-TEM can also detect morphological changes upon the utilization of different parameters for biomaterials formulation, such as the lipid–polymer ratio, the loaded agents, and the preparation method, determining the ratio between cubic particles and vesicles. By cryo-TEM, one can access the fast Fourier transform (FFT) motifs that correspond to an optical diffractogram, reflecting the structural symmetry, distances, and angles contained in the respective cryo-TEM images. In the case of internally structured nanoparticles without a periodic internal pattern, such as the sponges (*L_3_* mesophase), whose structure cannot be elucidated in detail by other methods, e.g., small angle X-ray diffraction, the cryo-TEM technique can provide a unique, detailed illustration of their morphology [28]. Cryo-field emission scanning electron microscopy (cryo-FESEM) can also be used to obtain new 3D information about the structure of cubosomes, by using sublimation of the surface water at −90 °C and coating with platinum [29]. Biological synchrotron small-angle X-ray scattering (BioSAXS), a special subcategory of SAXS, provides structural data from weakly scattering biological solutions in real time, accessing the internal nanostructure; the shape and the structural evolution of various kinds of proteins, peptides, and nucleic acids, such as siRNA, and lipid–protein assemblies; and their effect as encapsulated agents to the internal organization of the nanostructures [13,25].

Liquid crystals exhibit anisotropic properties and produce bright double refraction or birefringence when viewed between two light polarizers arranged in a crossed position, in contrast to the isotropic materials, which appear dark. Therefore, crossed polarized light microscopy (CPLM) is able to detect an anisotropic material but is restricted to micron or submicron particle dimensions such as the respective in colloidal liquid crystalline nanoparticles that require the use of cryo-TEM [26].

The size and the size distribution, as well as the nanoparticle cargo reflected by the ζ-potential (mV), are three important physicochemical parameters also characterizing the physicochemical stability of the colloidal dispersion over time that should be evaluated during the formulation process of the liquid crystalline nanoparticles. Dynamic and electrophoretic light scattering (DLS and ELS) techniques are well-established methods, typically used for this purpose.

Static light scattering (SLS) is a rapid and inexpensive technique towards the morphological characterization of nanosystems because it can provide useful in situ information about the structure of colloidal nanoparticles in real conditions and in dispersion state, and moreover a rapid screening and a quantification of the changes in the morphology of nanoassemblies under different environmental conditions, as well as for different formulation parameters [30,31]. By using the SLS technique in conjunction with the DLS technique, and measuring in a wide angular range, the *R_g_/R_h_* ratio can be estimated in order to understand the structure of nanoparticles in dispersion. The *R_g_/R_h_* ratio of the non-lamellar liquid crystalline nanoparticles was first calculated by Chountoulesi et al. [9]. Moreover, by SLS, fractal analysis can be carried out, resulting in the calculation of the fractal dimension (*d_f_*), which represents a parameter of the quantification of the morphology. For example, Pippa et al. [32,33,34,35,36] also correlated the *d_f_* with the self-assembly and the morphology of liposomes composed of different lipids, as well as mixed polymer–lipid liposomal nanocarriers. In the case of non-lamellar liquid crystalline nanoparticles, the fractal dimension was first calculated by Chountoulesi et al. [30] and was also correlated with the Gaussian curvature changes of the nanostructures upon temperature and pH alterations.

Thermal analysis by using techniques such as differential scanning calorimetry (DSC) can highlight phase transition temperatures, enthalpy changes, and stability of the materials upon temperature variation. Most recently, the combination of microcalorimetry (mDSC) with high-resolution ultrasound (HR-US) and rheology was used for the first time for the comprehensive evaluation of the thermal behavior of non-lamellar liquid crystalline nanoparticles, prepared from GMO or PHYT lipids [30]. Last but not least, other parameters of the internal microenvironment apart from the internal nanostructure, such as the microfluidity and micropolarity, should also be investigated. For this purpose, fluorescence spectroscopy employing a hydrophobic probe, such as the pyrene, can be applied, as was first reported by Chountoulesi et al. [9].

### 3.3. The Influence of the Environmental and Formulation Parameters in the Liquid Crystalline Nanostructure

It is crucial to be noted that the self-assembled liquid crystalline nanostructure formed is strictly influenced by the environmental parameters, such as temperature [9,30,37,38]; the solvent conditions, such as the salt concentration/ionic strength [39,40], the pH value [30,38,41,42], and the presence of serum proteins [9,38,43,44]; and the interactions with cells [45]. Regarding temperature and pH, we discuss these parameters in more detail in the next paragraph on stimuli-responsive nanoparticles.

An example in the literature concerning the influence of ionic strength was described by Liu et al. [39], where the increase in the ionic surfactant content (the anionic surfactant sodium bis(2-ethylhexyl)sulfosuccinate (AOT) or the cationic surfactant, didodecyldimethylammonium bromide (DDAB)) into the formulation increased the curvature toward the hydrophobic region, resulting in the phase transition from cubic to lamellar phase, and contrariwise the increase in the ionic strength of the environment (metallic ions) decreased repulsion between the headgroups of the ionic surfactant, resulting in a phase transition from lamellar to cubic phase, confirmed by using SAXS and cryo-TEM. Awad et al. [40] added the negatively charged dioleoylphosphatidylglycerol (DSPG) in monoolein, resulting in lamellar phases that were successfully transformed into cubic phases of either *Im3m* or *Pn3m* symmetry via Ca^2+^ cation-induced changes in the overall surface charge density.

Azmi et al. [43] investigated the interactions of PHYT-based non-lamellar crystalline dispersions with the plasma components and observed a decrease of the particle size, due to structural transition from the biphasic phase (*Pn3m* cubic coexisting with a hexagonal (*H_II_*) phase) to a neat hexagonal (*H_II_*) phase. Similarly, Chountoulesi et al. [9] observed acute size decrease of GMO and PHYT prepared liquid crystalline nanoparticles, stabilized by Poloxamer P407, after being incubated in fetal bovine serum containing albumin.

Tan et al. [45] carried out a bio-SAXS study, combined with a biological cell flow-through system, in order to monitor the effects of human blood circulating cells on the phase behavior of phytantriol-based liquid crystalline nanosystems. According to the results of this study, the nanosystems exhibited a time-dependent phase evolution after having been incubated with human monocytic cells (THP-1) in suspension.

The formulation parameters, such as the type of the amphiphilic molecule, the stabilizer concentration, the water content, and the presence of additives, play a crucial role on the conformation or the internal organization of LLCN. For example, the stabilizer Poloxamer P407 causes a double-diamond type (*Pn3m*) to primitive type (*Im3m*) phase transition only in GMO-prepared cubosomes, but not in PHYT ones, because the polymer is simply adsorbed at the surface of PHYT-prepared cubosomes and not partitioned, as in the GMO ones [46]. An increase in Poloxamer P407 concentration in the formulation can result in an increase of vesicles versus the non-lamellar structures percentage [9,24]. Furthermore, the type of the lipids, of the steric stabilizers, the nanoparticle surface charge, and the internal nanostructure (or lipid phase behavior), are all considered to be crucial factors affecting the cell–nanoparticle tolerability. For example, bicontinuous cubic mesophases are found to induce relatively higher cytotoxic and hemolytic effects than the more negatively curved inverse hexagonal and lamellar phase analogues across different cell lines and red blood cells of different species. This toxicity trend can possibly be correlated with the viscoelastic properties, while cubic is more “stiff” than hexagonal ones. Regarding the used lipid, PHYT, for example, is aggressive toward disrupting the cell nuclei membrane and blocking the plasma membrane turnover, which is an essential property for macropinocytotic activities [47]. Hinton et al. [16] also investigated the different cell-particles interactions, depending on the used lipid. In detail, the toxicity of phytantriol cubosomes is considerably greater than that of GMO cubosomes, because phytantriol cubosomes possess a greater ability to disrupt the cellular membrane (hemolytic activity) and cause oxidative stress.

Moreover, a transition of the internal mesophase conformation and of the nanostructure can be caused by the addition of other lipids or non-lipid additives. For example, the addition of vitamin E acetate in small amounts in the phytantriol–water system suppresses the temperature of the *Q_II_*-to-*H_II_*-to-*L_2_* phase transitions [48]. In a nanodispersion prepared by GMO/Vitamin E mixtures, the incorporation of the novel stabilizer d-α-tocopheryl poly(ethylene glycol)_2000_ succinate (TPGS-PEG2000) in specific concentrations resulted to a biphasic structure of coexisting inverse hexagonal (*H_2_*) phase and inverse discontinuous (micellar) cubic phase of the symmetry *Fd3m*, as well as causing the presence of “flower-like” vesicles, covering the non-lamellar LLCN, which were not ever observed in the dispersions stabilized by Pluronic F-127 [49].

The solvents used, such as ethanol, can also be considered as additives. Chen et al. [50] developed an injectable in situ liquid crystal formulation for intra-articular administration, in the form of viscous gel with long-term release of sinomenine hydrochloride upon water absorption. The addition of 5% vitamin E acetate yielded an in situ hexagonal liquid crystal being able to sustain the drug release for more than 10 days and was suitable for intra-articular injection for the treatment of rheumatoid arthritis.

The entrance of the drug molecule can also differentiate the internal nanostructure. Dimodan U/J (DU) is a monoglyceride mix of monolinolein and monoolein, being able to form a *Pn3m* cubic mesophase in excess of water. When Mionić Ebersold et al. [51] incorporated the antifungal lipophilic agent undecylenic acid, an increase of the negative curvature was caused due to an alteration of the lipid molecular packing, yielding a transformation from cubic phase towards the formation of hexosomes, as identified by the presence of curved striations (Figure 3c) or hexagonal periodicity (Figure 3b).

## 4. Non-Lamellar Lyotropic Liquid Crystals as Drug Delivery Nanosystems

Lyotropic liquid crystals, focusing on the non-lamellar ones, have gained increased research interest in the field of pharmaceutics, being ideal as therapeutic drug delivery nanosystems. They can be used as protein, peptide, and nucleic acid delivery nanosystems, and are administrated either as colloidal nanoparticles or as bulk phases. Their utility to carry diagnostic probes for biomedical imaging has also been well investigated. Some of their most important advantages that make them ideal drug delivery nanosystems are the following: great structural variety, nanostructure versatility, high grade of internal organization, and tunable morphological characteristics. Due to these advantages, they are able to carry great volumes of cargo, even greater than liposomes. They can also be loaded with different kinds of agents such as amphiphilic, hydrophobic, and hydrophilic agents, also providing sustained release of their content. Colloidal dispersions of cubosomes and hexosomes possess low viscosity that ensures flexible preparation and handling, particularly in the engineering of parenteral dosage forms. These carriers can also be easily functionalized through an array of surface engineering strategies. Moreover, they are characterized by great bioavailability and lower toxicity in comparison to other systems. Some basic invaluable properties that they provide as therapeutic nanosystems are the following: (1) improving drug bioavailability and reducing drug toxicity, (2) enhancing the stability of drugs, (3) exhibiting sustained or controlled drug release, (4) increasing the penetration of drugs, and (5) providing efficient solubilization of poorly water-soluble drugs. There are many different routes of administration, varying from oral, intravenous, intraperitoneal, ophthalmic, subcutaneous, intra-periodontal, topical, to transdermal, targeting a diverse range of sites, including internal organs, brain, eyes, and skin [1,7,15,52,53,54].

### 4.1. Controlling Drug Release Kinetic

A significant advantage of the liquid crystalline nanosystems is their ability to control the release kinetics of the contained drug. For example, Lee et al. [55] studied the in vitro sustained release behavior of a number of model hydrophilic drugs with various molecular weights. According to the obtained results, the cumulative amount of drug diffusion through the matrix followed a linear relationship with the square root of time, which represented a Higuchi diffusion controlled release profile, as was also previously confirmed by Boyd et al. [56] by testing a series of model hydrophobic and hydrophilic drugs (paclitaxel, irinotecan, glucose, histidine, and octreotide).

Generally, hydrophilic drugs are located close to the polar head or in the water channels, while hydrophobic drugs are loaded in the lipid layer, and amphiphilic drugs at the interface. For example, Esposito et al. [57] studied the performance of cubosomes as sustained percutaneous delivery systems with the model hydrophobic drug indomethacin. A prolonged release of indomethacin was observed by the cubosomes, thus exhibiting a significant long-lasting anti-inflammatory activity. The authors suggested that the content GMO lipid interacted with the stratum corneum lipids, causing the formation of a cubosomes depot in stratum corneum, from which the indomethacin was released in a controlled manner. The release of hydrophobic drugs is primarily controlled by the partition coefficient of the drug, its diffusion into the lipid bilayer, and its diffusion into the surrounded aqueous environment. For example, when Clogston et al. [58] increased the hydrophobicity of hydrophilic model drug tryptophan by alkylation, they observed a delayed release from GMO cubic phase that was attributed to an increase of its partitioning within the lipid layer.

Contrariwise, the release of hydrophilic molecules being solubilized within the aqueous domains depends more on the mesophase internal structure and the liquid crystalline topologies. The release of the entrapped molecules is supposed to be controlled by the unique microstructure of the cubic phase, where the active compound has to diffuse through the highly tortuous porous morphology, with different pore size and tortuosity of the water channels to access the external solution [59,60]. As Kulkarni et al. [59] describe, cubic phases of *Pn3m* type exhibit four aqueous channels (meeting at a tetrahedral angle), whereas the *Im3m* type consists of six aqueous channels (meeting at right angles) of larger size, yielding a rather rapid release. Between reversed micellar cubic phase (*Fd3m*) and bicontinuous cubic phase (*Pn3m*), in the case of the micellar one, the hydrophilic molecules need to systematically transport/cross a large number of lipophilic domains, while in the bicontinuous cubic phases, the molecules are diffused without crossing the lipid bilayer, resulting in a faster release rate [61,62]. Between hexagonal *H_II_* phase and bicontinuous cubic phase, there is a slower release in *H_II_* than the cubic phase, due to *H_II_* smaller pore size and reduced overall surface area, compared to the open nanostructured cubic phase [41,63]. Summarizing the correlation of the internal symmetry of the mesophase to the drug diffusion rate and thus the release rate: D_lamellar phase_ > D_cubic phase_ > D_hexagonal phase_ > D_micellar cubic_ [55,64]. We should mention that also in the case of poor water-soluble drugs, the internal nanostructure can affect their release kinetics. For example, Boyd [65] described the “burst” release of hydrophobic model drugs from bicontinuous cubic phase, due to its increased surface area and its nanostructure consisting of open water channels.

Taking into account all the above-described information, liquid crystalline nanostructures provide several opportunities towards the control of the drug diffusion rate by manipulating their internal architecture. Starting from the utilization of ionic interactions between the carried drug and the liquid crystalline carrier, such interactions can be created by the insertion of charged molecules within the mesophase. Anionic phospholipids, such as distearoyl phosphatidylglycerol (DSPG) [40] and dioleyl phosphatidylglycerol (DOPG) [66], or cationic lipids, such as 1,2-dioleoyl-3-trimethylammonium- propane (DOTAP) [67,68] and didodecyldimethylammonium bromide (DDABr) [69], have been reported that can trigger the formation of salt-induced non-lamellar nanostructures by interacting with surrounding ions. By employing pore-forming membrane proteins, for example, the bacterial pore-forming protein outer membrane protein F (OmpF) [70], unique topological interconnectivities between the aqueous nanochannels can be facilitated, significantly enhancing mesophase transport properties from the newly opened communication pathways.

Finally, the modification of the water channel size allows for the modulation of the diffusion rate. The swelling of the systems can be induced by adding varying concentrations of ionic lipids, such as DSPG [71], or hydration enhancers, such as octylglucoside [72] and sucrose stearate [73]. For example, Barriga et al. [74] created a library of highly swollen cubosome dispersions formed from ternary mixtures including monoolein, cholesterol, and an anionic lipid component. The authors also concluded that by changing the lipid type and the percentage of the incorporated anionic lipid, the lattice parameter and pore sizes can be precisely controlled, enabling opportunities of encapsulation and protection of biomolecules, as well as development of confined interfacial reaction environments. Sarkar et al. [75] investigated the phase behavior of quaternary lipid−water systems consisting of three different lipids (monoolein−cholesterol−phospholipid) and water, creating a large library of lipidic materials with bilayer structures, which mimic the native cell membrane more effectively and own significantly increased tunability, on the basis of nanostructural parameters, such as lattice parameter, aqueous channel size, and bilayer thickness. More specifically, phospholipids having phosphatidyl choline (PC), phosphatidyl ethanolamine (PE), and phosphatidyl serine (PS) head groups with a range of physiologically relevant saturated chain lengths from C12 to C18, as well as the singly unsaturated (C18:1), were investigated. According to the obtained results, the library contained also several extremely swollen cubic phases, being beneficial towards the successful encapsulation of large macromolecules such as proteins or nucleic acids. Recently, Zabara et al. [76] combined monoacylglycerols and phospholipids to design thermodynamically stable ultra-swollen bicontinuous cubic phases, with water channels five times larger than traditional lipidic mesophases, which exhibited re-entrant behavior upon increasing hydration. They utilized these phases in order to crystallize membrane proteins with small extracellular domains (ECDs), demonstrating the methodology on the Gloeobacter ligand-gated ion channel (GLIC) protein and overcoming the limitations of conventional in meso crystallization. We should mention that the control of drug release kinetics from liquid crystalline nanostructures can also be modulated by environmental stimuli, such as the temperature and the pH. The category of stimuli-responsive liquid crystalline nanostructures is analyzed in detail in Section 5 of the present review and thereby it is not further described here.

### 4.2. Non-Lamellar Lyotropic Liquid Crystalline Nanosystems (LLCN) for Anticancer Therapy

There are various examples in the literature regarding the successful application of liquid crystalline nanosystems in anticancer therapy. There are several examples in the literature describing the intravenous or subcutaneous in vivo administration of different anti-tumor drugs, such as docetaxel [77], paclitaxel [78,79,80], and 5-fluorouracil [81]. Other cases of anticancer therapies being administrated by other routes are also described in the following paragraphs. Various examples have also been reported where liquid crystalline nanocarriers provided promising anti-cancer effects, after being administrated in vitro in cancer cell lines, such as human carcinoma cell line HeLa [82], glioblastoma T98G cells [54], human hepatocellular carcinoma (HepG2) cells [83], and human breast cancer MDA-MB-231 cells [84].

Another reported strategy to decrease release rates and attain sustained release during anticancer therapy is the formation of liquid crystalline nanoparticles from amphiphiles that are themselves pro-drugs, acting simultaneously as nanocontainers and active agents. Gong et al. [85] worked with amphiphilic pro-drugs of 5-fluoroacil (5-FU) against breast cancer and other solid tumors. For example, amphiphilic prodrugs consisting of C18-alkyl derivatives of 5-fluorocytosine, with different numbers of unsaturations on the C18-alkyl chain, have been reported [85]. In a similar case, a phytanyl derivative of 5-fluorocytosine, which can also undergo enzymatic hydrolysis towards 5-fluorouracyl, has been studied to form nanostructured LLC particles [86].

### 4.3. Non-Lamellar LLCN Improving Oral Bioavailability

Liquid crystalline dispersions and especially the cubic ones are considered to exhibit significant advantages towards the oral delivery because their structure can protect the drug against degradation in the gastrointestinal (GI) tract. In the physiological environment of the GI tract, liquid crystals consist of an oil phase and a solubilized micellar phase during lipid digestion, which enhances drug solubility and bioavailability in the lumen. Another common advantage of liquid crystals is the reduction in systemic toxicity of high-toxicity drugs, such as the antibiotics and chemotherapeutics, thus improving therapeutic efficacy. Moreover, they increase the possibility of the drug to penetrate across the endothelial cell membrane, exhibiting improved uptake mechanisms, while their large hydrophilic surface, due to the presence of the water channels, allows for easy contact with the endothelial cell layer and can cross the water layer. Their internal nanostructure increases the absorption of hydrophilic drugs because it enhances their bioavailability by prolongation of gastric residence time, as well as increasing the absorption of hydrophobic drugs due to the increased stability and improved membrane penetration [26,87].

Nguyen et al. [88] demonstrated for the first time the ability of nanostructured liquid crystalline particles to maintain the absorption of a poorly water-soluble drug after oral administration in vivo. Moreover, they highlighted the effect of the different lipid (GMO or PHYT) of the formulation on the resultant pharmacokinetic profile. In detail, cubosomes formed from PHYT were shown to maintain the absorption of cinnarizine beyond 48 h after oral administration to rats, as depicted by the plasma concentrations. Contrariwise, cubosomes prepared by the digestible GMO did not sustain the absorption of the drug, leading to decreasing concentrations after 24 h. As a result, there was a significant enhancement in oral bioavailability from PHYT nanoparticles, compared to a cinnarizine suspension and oleic acid emulsion. The authors stated that the prolonged retention of the PHYT cubosomes in the stomach is attributed at least in part to the non-digestible nature of the lipid and the retaining of the cubosome structure. In conclusion, the potential use of non-digestible liquid crystalline nanostructured particles creates opportunities for novel sustained oral drug delivery systems.

Mechanistically, liquid crystalline nanoparticles are known to increase the oral bioavailability by virtue of bioadhesiveness, membrane fusing properties, clathrin/caveolae/lipid raft-mediated endocytosis, and superior encapsulation and solubilization. It has been reported that significant enhancement of the oral bioavailability of paclitaxel can be achieved by the encapsulation of paclitaxel inside LCNPs, which would not only help the drug to escape recognition by P-glycoprotein but would also improve intestinal epithelial permeability by increasing membrane fluidity. Namely, the in vivo pharmacokinetic study showed that the oral bioavailability of paclitaxel-loaded liquid crystalline nanoparticles was 2.1 times that of the Taxol^®^ [80].

There are many other examples in the literature on liquid crystalline nanosystems, either bulk mesophases or nanoparticles, being studied for the oral administration of insulin [89], other peptides such as cyclosporine A [90], anti-cancer treatments [91], anti-hypertensive treatments [92], and antibiotics [93], as well as oral vaccination with Ovalbumin and Quil-A adjuvant [94]. Swarnakar et al. [95] described the enhancement of antitumor efficacy, improving bioavailability and safety of orally co-administered doxorubicin with the antioxidant coenzyme Q10 by liquid crystalline nanoparticles. The co-administration of CoQ10-liquid crystalline nanoparticles with doxorubicin–liquid crystalline nanoparticles completely abolished doxorubicin-induced cardiotoxicity, as confirmed by the levels of all biochemical parameters.

Most recently, a liquid taste-masking system based on lyotropic liquid crystalline nanoparticles was developed by Fan et al. [96]. Cefpodoxime proxetil, a typically bitter drug used as an antibiotic in pediatric medicine, was encapsulated into the nanoparticles as an effort to improve its taste. The formulation exhibited a sustained-release profile well fitted to the Higuchi model, indicating that diffusion and erosion were both responsible for the drug release, while the desired taste-masking ability was confirmed by electronic tongue and compared to free Cefpodoxime proxetil and commercial product. The authors concluded that the liquid crystalline formulation showed a great potential for pediatric oral delivery, being able to improve the compliance of pediatric patients by masking the bitterness of API and reducing the dosing frequency by 24 h sustained release.

### 4.4. Non-Lamellar LLCN for Skin Administration

The topical administration of drugs with liquid crystals, for example though the skin, can be successfully utilized, because liquid crystals can localize the drug within the stratum corneum and improve drug penetration by increased transdermal permeability. The cubic phase may interact with the stratum corneum structure, leading to the formation of a cubosomal mixture from GMO and the native lipids of stratum corneum that acts as a cubosome depot, where a controlled release has taken place. GMO can interact with the stratum corneum, being actually an absorption enhancer that boosts the intercellular lipid fluidity. The cubic phase can also form a biological membrane-like structure with a strong bioadhesive property to the skin, while it also exhibits the proper viscosity and mucoadhesiveness for topical applications. Apart from the liquid crystalline bulk phases that possess proper viscosity and mucoadhesiveness for topical applications, the liquid crystalline nanoparticles are also used due to their greater ease of handling, reduced viscosity, and the ability to deliver higher drug payloads than the bulk analogues. Tissue hydration is an extra advantage of the water included in the liquid crystalline nanostructure [26,97,98].

There are many examples in the literature describing liquid crystalline formulations for dermal treatment for burns [99,100]; topical delivery of antimicrobial peptides for treatment of skin infections caused by bacteria, such as *Staphylococcus aureus* [101]; topical photodynamic therapy [102]; therapy of melanoma [103,104]; rheumatoid arthritis [105]; atopic dermatitis [106]; and hair loss [107]. Apart from drug molecule dermal delivery of peptides [108], as well as siRNAs [109], have been reported. For example, GMO-based hexosomes with cationic charge, for complexing with siRNA, showed higher penetration into the skin without causing skin irritation [109].

Liquid crystalline nanosystems being topically administrated can also be formulated in hydrogels. For example, Thakkar et al. [100] developed cubosomal hydrogels (cubogels), loaded with silver sulfadiazine and aloe vera for topical treatment of infected burns, with a higher burn healing rate than the corresponding marketed product as depicted by in vivo studies, useful against deep second-degree burns.

Topical liquid crystalline formulations exhibit less cytotoxic effects with higher therapeutic efficacy, which is desirable in antibiotics and nonsteroidal anti-inflammatory drug (NSAID) therapies. One example in the literature was the development of liquid crystalline nanosystems of cubic phase incorporating celecoxib. In this case, the formulation was applied topically, and the drug was localized within the stratum corneum [110]. Ringing gels are a cubic liquid crystalline-based system with a high concentration of surfactants, increasing the permeability through the stratum corneum, being commercially available and used as topical NSAID formulations. Typical examples are Dolgit Mikrogel (ibuprofen) and Contrheuma Gel Forte N (bornyl salicylate, ethyl salicylate, and methyl nicotinate) ringing gels [111].

A severe pathological skin condition where high permeability of drugs is required in order to achieve efficient therapeutic effects is melanoma. Yu et al. [103] prepared cubic phase from GMO loaded with mitoxantrone as a transdermal formulation for the convenient melanoma therapy. The transdermal permeability of mitoxantrone was higher by the cubic phase compared to that of the mitoxantrone solution because the unique internal structure of the cubic phases can deliver the hydrophilic drug, crossing through the skin and penetrating the melanoma tissue.

### 4.5. Non-Lamellar LLCN for Ocular, Brain, and Pulmonary Delivery

Non-lamellar liquid crystalline nanosystems have also been investigated for ocular topical delivery. Eye drops exhibit key problems including corneal permeability, retention times, and low solubility of some drugs, resulting in poor drug bioavailability that liquid crystalline formulations overcome. Several cases in the literature report liquid crystalline nanosystems formulated as eye drops, being in vivo tested in rabbit models, delivering anti-inflammatory drugs, such as dexamethazone [112], flubiprofen [113], and glaucoma treatment such as pilocarpine [114] and brinzolamide [115]. For example, cyclosporine A was incorporated in GMO nanoparticles, stabilized by Poloxamer P407, resulting in a decreased ocular irritancy and improved corneal penetration, when compared with a control cyclosporine A formulation [116]. Liu et al. [117] developed a liquid crystalline nanosystem, owning a *H_II_* nanostructure, prepared by GMO, wherein they incorporated the absorption enhancer Gelucire 44/14 and octadecyl-quaternized carboxymethyl chitosan adjuvants for the ocular delivery of tetrandrine, an agent against chronic keratitis, cataracts, retinopathy, and glaucoma, which exhibited enhanced transcorneal penetration of tetrandrine in a rabbit model. Similarly, GMO cubosomes of *Pn3m* internal symmetry stabilized with Poloxamer P407 and loaded with Timolol maleate, a beta blocker commonly used to treat glaucoma, exhibited higher penetration ex vivo than the commercially available product, increased retention times in vivo, and had an enhanced ocular pressure lowering effect, while neither cytotoxicity nor histological impairment in the rabbit corneas were observed [118]. More recently, Alharbi et al. [119] developed a ciprofloxacin-cubosomal in situ gel in order to improve eye permeation, prolong the ocular retention time, and enhance the antimicrobial activity of the antibiotic, compared with commercial drops.

The fact that liquid crystalline nanoparticles can incorporate surfactants, such as polysorbates (for example Tween 80), make them also suitable for drug delivery to the brain. The literature suggests that after parenteral administration, the plasma lipoproteins ApoB and ApoE adsorb Tween 80-coated nanoparticles to form a protein corona, which can then mimic the endogenous transport of low density lipoproteins bound to ApoB or ApoE, across the BBB via receptor-mediated transcytosis. Thus, Tween is considered to be an ideal stabilizer for the surface coating of cubosomes targeting the brain. From a nanotechological point of view, the addition of Tween 80 to mixtures of phytantriol and water can stabilize cubosomes [120]. Abdelrahman et al. [121] enhanced the risperidone delivery to the brain through the transnasal route via optimization of cubosomal gel. Cubosomes were prepared from GMO lipid and were stabilized by Poloxamer P407 and Tween 80. The optimal cubosomal formulation exhibited significantly higher transnasal permeation and brain distribution, being ideal for brain targeting through the transnasal route. Elnaggar et al. [122] incorporated Tween 80 in liquid crystalline formulations for the brain-targeted oral delivery of piperine. The prepared cubosomes were able to significantly sustain drug in vitro release, while the in vivo studies revealed a significantly enhanced piperine cognitive effect and even a restored cognitive function to the normal level at rats with sporadic dementia of Alzheimer’s type, indicating the potential of the proposed nano-formulation to stop Alzheimer disease progression. Proteins related with treatment of central nervous system diseases, which must cross the brain–blood barrier, can also be delivered by liquid crystalline nanoparticles. For example, odorranalectin surface-decorated cubosomes encapsulating the anti-Alzheimer S14G-HN peptide, a very promising anti-Alzheimer peptide, have been developed, and according to the results, showed enhanced therapeutic effect via intranasal administration, as well as sustained peptide release [123].

Liquid crystalline phases were proven to be suitable for the engineering of lactose crystals and to be utilized as efficient carriers in dry powder inhalation (DPI) formulations. In particular, saturated lactose solution was poured in molten GMO lipid, which subsequently was transformed into gel. The aerosolization parameters of the particles, determined by using salbutamol sulfate as a model drug, were equivalent to Respitose^®^ ML001 and thus are suitable to be used as carrier in DPI formulations [124].

### 4.6. Injectable Non-Lamellar Liquid Crystalline Depot Systems for Sustained Delivery

Sustained-release injections are designed to release a drug substance at a predetermined rate to maintain its effective plasma concentration for months. Although injectable polymeric microspheres (for example, from poly(lactic-co-glycolic acid (PLGA)) or implants have been developed as injectable sustained release systems in various studies, they are difficult to be prepared and can reduce the stability of protein drugs. These disadvantages can be overcome by injectable liquid crystal-forming systems (LCFS). The liquid crystalline mesophase is spontaneously formed from the LCFS in an aqueous fluid. The formed tortuous networks of their aqueous nano-channels are able to sustain the drug release. For example, for the efficient therapy of hepatitis B using entecavir, the drug must be taken consistently every day, due to disease re-occurrence in cases of discontinuation. Kim et al. [125] developed a novel LCFS of hexagonal phase for the sustained delivery of entecavir. A pharmacokinetic study in rats was carried out, showing sustained release of entecavir for 3–5 days from LCFS formulation. In another study [126], LCFS containing sorbitan monooleate (SMO) was investigated for sustained release injections of leuprolide acetate. The LCFS formed the hexagonal liquid crystalline phase. Both in vitro release test and in vivo pharmacokinetic and pharmacodynamic studies showed a sustained release of leuprolide. When compared with a commercial depot formulation of leuprolide, the LCFS exhibited a significantly reduced initial burst with sufficient suppression of testosterone. Later, Báez-Santos et al. [127] showed that tocopherol acetate can play a major role in mitigating drug release by altering the physicochemical properties of the liquid crystalline matrix, indicating the use of tocopherol acetate as a tailoring agent. More specifically, formulations with low amounts of tocopherol acetate and higher water uptake capacities had a higher propensity towards erodibility and thus in vivo biodegradability.

Yang et al. [128] developed an injectable in situ liquid crystal formulation for the local delivery of minocycline hydrochloride, being administered as a periodontal pocket topical delivery system, for chronic periodontitis treatment. Precursor formulations were prepared by PHYT with propylene glycol (PG) in specific ratios (Figure 4a), until the formulation was optimized and loaded with the drug to an injectable formulation. The in vitro release experiments showed that the minocycline hydrochloride-loaded in situ cubic liquid crystalline formulation presented higher cumulative and sustained releases for 4 days in comparison with Periocline^®^ (Figure 4b), while therapeutic effects on periodontitis were also obtained.

All the above-described drug delivery applications of the LLCN are summarized in Table 1.

### 4.7. Non-Lamellar LLCN as Vaccines

Cubosomes and hexosomes have been referred to as vaccine adjuvants that carry both immune enhancers and antigens to regulate the immune response of the body. It has already been reported that nonlamellar structures (e.g., cubic and hexagonal) show fusogenic properties that are able to deliver antigens directly to the cytosol of antigen-presenting cells (APCs), stimulating cytotoxic T lymphocyte (CTL) immune responses. Most protein antigens are negatively charged at neutral pH, as are cubosomes. To link antigens on cubosomes, a cationic surfactant is usually utilized.

Moreover, the supramolecular structure of the lyotropic liquid crystalline phase is considered to influence the immunostimulatory activity of lipid-based nanocarriers. Rodrigues et al. [129] designed PHYT hexosomes with the immunopotentiator monomycoloyl glycerol-1 (MMG-1). The effect of the nanostructure on the adjuvant activity was studied by comparing the immunogenicity of phytantriol/MMG-1 hexosomes with MMG-1-containing liposomes in mice. According to the results, the MMG-1-based hexosomes potentiated significantly superior MOMP-specific humoral responses in comparison with liposomes. The authors suggested that hexosomes exhibit great adjuvant potential, and engineering of the supramolecular structure can be used to design adjuvants with customized immunological properties. Another novel self-adjuvanting hexosome-based PHYT nanocarrier with mannide monooleate (MaMo), which is an emulsifier applied in several adjuvant systems, was developed by Rodrigues et al. [130]. A bulk phase composition of phytantriol/MaMo (14 wt %) showed hexagonal (*H_II_*) phase over body temperature, and subsequently hexosome nanoparticles were stabilized with different concentrations of either poloxamer 407, Myrj 59, or Pluronic F108. The nanosystems maintained their *H_II_* structure during the modification with either positively or negatively charged lipids and the loading with model antigens, indicating remarkable structural robustness and demonstrating that they can be utilized as antigen delivery carriers. In another work, an antigen against the helminth parasite *Fasciola hepatica*, namely, *F. hepatica* Kunitz-type molecule (FhKTM), was formulated with a liquid crystal nanostructure formed by self-assembly of 6-O-ascorbyl palmitate ester (Coa-ASC16) and the synthetic oligodeoxynucleotide containing unmethylated cytosine-guanine motifs (CpG-ODN). According to the results, the immunization of mice with FhKTM/CpG-ODN/Coa-ASC16 induces protection against *F. hepatica* infection by preventing liver damage and improving survival after *F. hepatica* infection. The vaccination-induced IL-17A blockade during infection decreased IgG2a and IgA antibody levels as well as IFN-g production, leading to an increase in mortality of vaccinated mice [131].

The fact that liquid crystalline nanosystems can be easily surface functionalized increases their adjuvanticity potential. For example, Qiu et al. [132] designed chitosan (CS)-modified ginseng stem-leaf saponin (GSLS)-encapsulating cubosomes (Cub-GSLSCS), while GSLS have been widely used as immune-modulators and pharmaceuticals. The Cub-GSLSCS showed excellent storage stability and sustained OVA release for up to 28 days, improving immunopotentiation, cellular uptake, and stimulated cytokine secretion. According to in vivo results in mice, the ratio of CD4^+^/CD8^+^ T lymphocytes was increased, and dramatically high OVA-specific IgG, IgG1, and IgG2a levels and stimulated secretion of cytokines were induced. Cub-GSLSCS may be a potential vaccine delivery system with long-term sustained immunogenicity by inducing both humoral and cell-mediated immune responses. Another immunostimulant polysaccharide (PS), extracted from *Ganoderma lucidum*, a promising adjuvant for vaccines, was incorporated into cubosomes to generate PS–cubosome (Cub-PS) nanoparticles. The results demonstrated that Cub-PS elicited more potent immune responses than Cub or PS alone. The enhanced immune responses might be attributed to the promotion of antigen transport into draining lymph nodes and efficient dendritic cell activation and memory T-helper cell differentiation in draining lymph nodes. These results demonstrate that cubosomes can enhance the adjuvant activity of immunostimulants [133]. Moreover, in the study of Liu et al. [134], cetyltrimethylammonium bromide (CTAB) and poly (diallydimethy ammoniumchloride) (PDDAC) were linked directly or indirectly on the surface of *Ganoderma lucidum* polysaccharide cubosomes (GLPC). Furthermore, ovalbumin (OVA) antigens were adsorbed on CTAB-GLPC and PDDAC-GLPC nanoparticles. The observed enhanced humoral and cellular immune responses of PDDAC-GLPC-OVA were attributed to the maturation of dendritic cells into draining lymph nodes, activation of the spleen, and secretion of cytokines into systemic circulation, being a potential adjuvant for protein–antigen vaccines.

In order to link antigens on cubosomes, a cationic surfactant is usually selected to provide a positive charge. For example, polygonatum sibiricum polysaccharide (PSP), of *Polygonatum sibiricum*, is an immunostimulant to improve immune responses. Liu et al. [135] investigated the immunomodulation effects of ovalbumin (OVA)-absorbed cetyltrimethylammonium bromide-modified *Polygonatum sibiricum* polysaccharide cubosomes (CTAB-modified PSP-Cubs/OVA). As the results showed, CTAB-modified PSP-Cubs/OVA could promote the production of OVA-specific IgG in serum, increase the ratio of CD4^+^ to CD8^+^, significantly activate dendritic cells, and promote lymphocyte proliferation. The results also indicated that it could promote the secretion of related cytokines and the proliferation of lymphocytes, stimulate the cellular immune response, and increase the level of humoral immunity. Above all, CTAB-modified PSP-Cubs had good adjuvant activity, thus being able to serve as an effective vaccine adjuvant to induce immune responses.

Emerging needs for new, safe, and efficient vaccine nanoplatforms, as well as a great scientific interest, have arisen due to the coronavirus infectious disease-19 (COVID-19) pandemic, caused by the SARS-CoV-2 coronavirus [136]. Among the different market-approved or still candidate vaccines, there are many of them being formulated in nanotechnological platforms, in a liquid crystalline state, such as the lipid nanoparticles. For example, BioNTech/Pfizer and Moderna encapsulate their mRNA vaccines within lipid nanoparticles of lamellar form. Apart from the lipid-based nanoparticles, there is also another type of nanoparticle that is formulated by the saponin glycosides. The Novavax vaccine (NVX-CoV2373) is a subunit vaccine, based on a technology already used in a few approved products. It contains the spike protein genetic sequence of the original SARS-CoV-2 strain, derived from moth cells, and its Matrix-M™ adjuvant is based on a saponin extracted from the Chilean soapbark tree (*Quillaja saponaria*). Thus, NVX-CoV2373 is a MatrixM™-adjuvanted recombinant nanoparticle vaccine engineered form. The containing protein is recombinant SARS-CoV-2 (rSARS-CoV-2) protein, constructed from the full-length (i.e., including the transmembrane domain), wild-type SARS-CoV-2 spike glycoprotein [136,137], an adjuvant based on saponin extracted from the *Quillaja saponaria* Molina tree that induces high and long-lasting levels of broadly reacting antibodies supported by a balanced TH1 and TH2 type of response, including biologically active antibody isotypes, multifunctional T cells, and cytotoxic T lymphocytes, as well as promoting rapid and profound effects on cellular drainage to local lymph nodes, creating a milieu of activated cells including T cells, B cells, natural killer cells, neutrophils, monocytes, and dendritic cells. Matrix-M is composed of 40 nanometer particles based on saponin extracted from the *Quillaja saponaria* Molina bark together with cholesterol and phospholipid [137,138,139,140]. Matrix-M provides strong and long-lasting immune responses that can enable dose-sparing. The vaccine led to antibody neutralization titers in all patients both against the S protein (after one dose), as well as to the wild-type virus (after two doses). Vaxine announced positive safety data as their latest subunit vaccine has entered advanced clinical trials [136]. Importantly, the Matrix-M1 adjuvant was dose-sparing and induced CD4^+^ effector memory T-cell responses that were biased toward a Th1 phenotype, which may play a role in reducing the theoretical possibility of antibody-dependent enhancement (ADE) of SARS-CoV-2 infection. The fact that the formulation can be stored in refrigerators is a practical advantage that could boost distribution to low- and middle-income countries [137,139].

## 5. Development of Stimuli-Responsive Non-Lamellar LLCN

Stimuli-responsive nanosystems are designed to exploit in a “smart” way the altered conditions (for example temperature, pH, and enzyme concentration) that take place in pathological tissues, causing triggered content release in the targeted tissue and resulting in enhanced bioavailability, prolonged blood circulation time, and overall increased therapeutic efficacy. Although the P407 is a commercially available and biocompatible polymer, having been proven to be an excellent stabilizer, it lacks functional targeting groups or stimuli-responsive groups. Thus, the upgrade of the conventional liquid crystalline nanosystems to stimuli-responsive ones by adding the suitable biomaterials may elicit on-demand drug delivery [141].

### 5.1. pH-Responsive Non-Lamellar LLCN

pH-responsive systems are able to exploit well-characterized pH differences inside the human body. pH differences, where the pH-responsive nanosystems usually target, exist between normal blood and pathological tissues (e.g., due to infection, inflammation, and cancer tending to acidic pH), between certain intracellular compartments (i.e., cytosol, endosomes, and lysosomes), as well as along the gastrointestinal track. “Smart” molecules such as polymers, lipids, and peptides that are biocompatible and responsive to particular pH conditions, due to their functional ionizable groups, are utilized for this scope [142,143].

#### 5.1.1. pH-Responsive Non-Lamellar LLCN Employing pH-Responsive Polymers

In a recent example in the literature, Kluzek et al. [42] used a pH-sensitive polymer designed to strongly interact with the lipid structure at low pH, providing pH sensitivity to monoolein cubosomes and promoting a triggered drug release at acidic pH. The used polymer was the pseudopeptidic polymer poly(l-lysine-iso-phthalamide), designed to mimic the membrane penetrating peptides, exhibiting carboxylic groups that can promote conformational changes. More specifically, at neutral pH, the polymer exhibits extended charged polymeric chains, which are transformed to globular state at acidic conditions, promoting binding with the lipidic membrane and eventually causing its disruption, facilitating triggered drug release in the intracellular acidic conditions.

Kwon and Kim [144] triggered a pH-dependent release from MO cubic phase by a complex coacervation between alginate (HmAL) and silk fibroin (HmSF) in the water channels. The silk fibroin and alginate were hydrophobically modified in order to be immobilized in the water channels. Thus, the release can be controlled in a pH-dependent manner due to the complex coacervation between the protein and the negatively charged stearylamine groups of polysaccharide. Under acidic conditions, coacervate will block the water channels of cubic phase, suppressing the release. As the pH of release medium increased to neutral and alkali conditions, there was a higher release, possibly due to the dissolution of complex coacervation. Authors concluded that the cubic phase could be exploited as a pH-sensitive carrier for the oral delivery of an acid-labile drug.

Crisci et al. [145] synthesized a weak polyacid-lipid conjugate, namely, the poly- (acrylic acid) (PAA) with the 1,2-dimyristoyl-sn- glycero-3-phosphoethanolamine (DMPE) and incorporated it into a lipid mesophase. At low pH and in the absence of salt (NaCl), the neutral protonated PAA chains adopt a coil (brash) conformational state that leads to the formation of a swollen lamellar structure. Upon the addition of salt at low to intermediate pH values, two lamellar phases, one collapsed without a polymer and one expanded structure with a polymer, are coexisting. Finally, when the polymer is fully ionized (at alkaline pH), the extended conformation of the polymer generates a cubic phase, co-existing with the lamellar. The authors also concluded that the state of ionization of the polyelectrolytes and their pH-dependent conformation should be taken into account during the development of such self-assembled soft nanostructures.

#### 5.1.2. pH-Responsive Non-Lamellar LLCN Employing pH-Responsive Molecules (Lipid or Surfactant)

Negrini and Mezzenga [41] employed linoleic acid as the pH-sensitive triggering molecule in order to develop a food-grade pH-responsive lyotropic liquid crystal system. The developed system was able to switch in a reversible way both its structure and physical properties, resulting in pH-controlled release from a monolinolein cubic phase. By employing the SAXS technique, the authors managed to monitor the structural transition between the neutral pH (simulating intestine conditions) and the pH = 2 (simulating stomach conditions). The linoleic acid, being a weak fatty acid (pKa ≈ 5), is protonated at acidic pH, inducing structural changes from cubic phase at neutral pH to hexagonal phase in an acidic environment. Depending on its different structures, it exhibits different release kinetic profiles in the different pH conditions. The proposed system was able to retain the model hydrophilic drug phloroglucinol in the acidic human stomach and release the drug in neutral pH conditions, thereby being suitable for oral administration to intestine or colon tracts.

Negrini et al. [146] also proposed another liquid crystalline pH-responsive system with transition from hexagonal phase at higher pH to cubic phase at lower pH < 5.5, induced by the protonation of the incorporated weak amphiphilic base pyridinylmethyl linoleate taking place at acidic pH. Pyridinylmethyl linoleate is neutral at pH 7 and above but is protonated to positive under acidic pH conditions. This protonation induces changes in the lipid bilayer curvature. Due to the controlled anticancer drug doxorubicin release from the mesophases that was carried out in the low pH microenvironment of tumors, the system provoked human colon cancer cell death at an acidic pH, mimicking tumor condition.

Prajapati et al. [147] developed pH-sensitive nano-self-assemblies of the poorly water-soluble anticancer drug 2-hydroxyoleic acid (2OHOA), prepared from GMO. The pH-induced phase transformations were caused by the carboxylic group of 2OHOA, resulting in charge-repulsions at the lipid–water interface, confirmed by ζ-potential measurements. At pH < 4.0, the mostly protonated 2OHOA was inserted into the hydrophobic domains of the cubosomes towards the formation of the inverse hexagonal phase. The gradual deprotonation of the 2OHOA molecules upon increasing pH resulted in electrostatic repulsions among its deprotonated carboxylic head groups, causing the swelling of the cubic *Pn3m* and *Im3m* phases and eventually, at pH ≥ 4.5, to the cubosomes to vesicle modification. The authors concluded that these results are promising towards new pH-responsive anticancer nanocarriers for the targeted delivery of chemotherapeutics to the local microenvironment of tumor cells.

Li et al. [148] developed pH-responsive liquid crystalline lipid nanoparticles, from monoolein and oleic acid, which were dual-loaded by the phytochemical *Brucea javanica* oil (BJO) and the antitumor doxorubicin hydrochloride (DOX). The pH sensitivity of the nanocarriers is attributed to the oleic acid fraction of the lipid phase and the BJO ingredient, which contains unsaturated fatty acids. BJO is a traditional herbal medicine that strongly inhibits the proliferation and metastasis of various cancers. SAXS revealed a pH-induced inverted hexagonal (normal) to cubic (pH around cancer cells) to emulsified microemulsion (pH inside cancer cells) phase transition. The optimized DOX–BJO–LCNP formulation showed long-term stability, high encapsulation efficiency, and controlled drug release properties. Moreover, an enhanced antitumor effect, in comparison to the suspension of the free drugs, as well as a reversing the resistance of the MCF-7/DOX cells to DOX, were observed.

Ribeiro et al. [149], by adding 10.0% (*w*/*w*) of decyl betainate chloride (DBC), which is a cleavable cationic surfactant, into phytantriol/Pluronic-based dispersions, managed to gradually induce lamellar-to-cubic-to-hexagonal phase transitions, in response to neutral/alkaline pH. SAXS was used to monitor the mesophase transition upon the addition of DBC and pH variation. TEM revealed the presence of niosomes after the addition of DBC. The niosomes formed in these systems are pH-responsive with lamellar (niosomes)-to-reverse cubic-to-reverse hexagonal phase transitions in neutral and alkaline environments. Conversely, the formulations resisted phase changes in acidic environments. The authors concluded that there is a new perspective of oral applications, with potential pH-triggered release of drugs at the site of action under conditions that require gastro-resistance.

### 5.2. Thermoresponsive Non-Lamellar LLCN

The “smartness” of the thermoresponsive nanosystems is owned to the ability of their biomaterials to undergo temperature-dependent phase and conformational transitions that cause structural changes in the resultant systems, triggering the controlled release of their content. It should also be taken into account that pathological tissues (40–42 °C), such as tumors, exhibit elevated temperatures compared to physiological ones.

As for the lyotropic liquid crystalline nanosystems, their phase transition is strictly influenced by the temperature change and can be further monitored by accurate design of their formulation. The most commonly utilized GMO lipid has been proven to perform typical liquid crystal thermal expansivity, while the lattice parameter of all GMO mesophases decreases with increasing temperature [11,150]. By examining a similar lipid-forming non-lamellar liquid crystalline mesophase, namely, the monolinolein lipid (MLO), it was found that MLO particles have a cubic internal structure at 25 °C that is transformed to inverse hexagonal after heating at 58 °C and to inverse micellar phase upon further heating to 87 °C, in a reversible manner. MLO-based particles also expel water upon heating (deswelling/shrinkage) and take up water again upon cooling (swelling) also in a reversible way, termed as the “breathing mode” [151]. In the case of P407-stabilized PHYT particles, an inverse micellar solution is observed above 50 °C, while the cubic structure of *Pn3m* symmetry re-appears with cooling down [152].

After extensive structural studies, the phase diagrams of lipids such as MO and the effect of the temperature have been mapped [153,154]. By adding additive molecules, the temperature-responsiveness of the liquid crystalline nanosystems can be fine-tuned. Fong et al. [63] developed a thermo-responsive lipid-based liquid crystalline system, being the first in situ-triggered on-demand drug delivery system, by adding non-lipid molecules, such as vitamin E acetate and oleic acid. More specifically, two formulations were investigated, namely, phytantriol–vitamin E acetate and GMO–oleic acid. By using glucose as a model hydrophilic drug, the drug diffusion was shown to be reversible on switching between the *H_II_* and *Q_II_* nanostructures at temperatures above and below physiological temperature, respectively. Stimulated changes in drug release from the matrix were observed during in vivo proof of concept experiments, being subcutaneously administrated in rats also. A heat or cool pack at the injection site was applied in order to confirm their thermo-responsive ability, as was anticipated from in vitro release behavior. The authors commented in the promising utility of these systems as “on-demand” drug release delivery vehicles.

Barriga et al. [155] employed a combination of different lipids in order to trigger the thermo-responsiveness of monoolein-prepared liquid crystalline nanosystems. They combined monoolein, cholesterol, and the negatively charged phospholipids 1,2-dioleoyl-sn-glycero-3-phospho(10-rac-glycerol) (DOPG) and 1,2-dioleoyl-sn-glycero-3-phospho-l-serine (DOPS) in order to prepare highly swollen primitive (*Im3m*) symmetry bicontinuous cubic phases, highly sensitive to both temperature and pressure.

Dabkowska et al. [156] provided temperature responsiveness in lipid-based non-lamellar cubic lipid layers by embedding spherical, submicron-sized polymeric poly(N-isopropylacrylamide) (PNIPAM) nanoparticles. The polymeric nanoparticles were used as thermoresponsive controllers of the hydration of the liquid crystalline surface layers. PNIPAM exhibits limited solubility at elevated temperature, changing from the swollen to the collapsed state with subsequent release of water around 32 °C. Through this PNIPAM change, nanogels enable on-demand release of water while the lipid matrix remains intact. The authors concluded that the proposed systems can be further exploited as new switchable nanostructured materials for controlled release.

### 5.3. Dual Stimuli-Responsive (pH and Temperature) Non-Lamellar LLCN

More recently, Chountoulesi et al. [38] incorporated polycations of PDMAEMA for the first time in liquid crystalline nanoparticles. In this study, the stimuli-responsive amphiphilic block copolymer poly(2-(dimethylamino)ethylmethacrylate)-b-poly(laurylmethacrylate) (PDMAEMA-b-PLMA) was proposed as a novel stabilizer for liquid crystalline nanoparticles with extra advanced stimuli-responsive properties. High positive values of ζ-potential were observed, probably due to the charged amino groups of PDMAEMA in their structure. The morphology of the nanosystems was illustrated by cryo-TEM studies, revealing diversity of vesicles, organized cubic phases of *Im3m* internal symmetry, and sponges of medium-level internal organization (Figure 5). All nanosystems presented pH-induced alterations of their charge (Figure 5). More specifically, their positive charge was increased in a more acidic environment and was decreased at neutral pH, which is useful in pharmaceutical applications. However, alterations of *R_g_/R_h_* ratio values were observed in acidic pH, indicating a pH-induced structural re-arrangement. Taking into account the temperature-responsiveness of the PDMAEMA block, the response to temperature was also monitored. Irreversible physicochemical (size and size distribution) and morphological alterations (*R_g_/R_h_* ratio) were observed, probably due to the temperature-induced shrinkage of PDMAEMA by its conversion from being hydrophilic to more hydrophobic at increased temperatures. Thus, the resultant nanosystems were characterized as dual stimuli-responsive (both towards pH and temperature).

Recently, Fong et al. [37] carried out SAXS studies in temperature- and pH-sensitive cubic phases of monoolein individually incorporating five *cis* unsaturated fatty acids with hydrocarbon chain lengths between 18 and 24 carbons, namely, oleic acid, vaccenic acid, gondoic acid, erucic acid, or nervonic acid, that all possess wedge-shaped surfactant molecular geometry. The authors managed to establish their partial temperature-composition phase diagrams and structure, identified the presence of micellar cubosomes, and evaluated their pH-responsiveness in phosphate-buffered saline (PBS). The micellar *Fd3m* cubic phase was formed at pH around 4.9 and a very low ionic strength. The temperature, the fatty acid concentration, and the pH all directly impacted the formation and stability of *Fd3m* cubic phase. Finally, the authors concluded that the observed low-energy inverse micellar cubic-to-emulsion phase transformations in the monoolein with oleic acid and vaccenic acid systems at physiological temperatures may be advantageous for further pharmaceutical applications. The transformations were observed within a physiological temperature tolerance (30–45 °C) over a wide compositional window.

The above-described examples regarding the stimuli, along with the other environmental and formulation parameters referred to in previous (Section 3.3 and Section 4.1) that can influence the internal organization and the behavior of the LLCN, are summarized in Table 2.

## 6. Conclusions and Future Perspectives

Non-lamellar lyotropic liquid crystalline nanosystems have gained increasing interest recently in the pharmaceutical nanotechnology field. The present review focused on the cubic and hexagonal nanosystems, whose high level of internal organization provides great opportunities. The cubic and hexagonal nanostructures display high solubilization and encapsulation capacities for a variety of guest molecules, including drugs, peptides, proteins, and nucleic acids, as well as the ability to protect the active molecules, improve their bioavailability, and control their release kinetics profile. According to a plethora of literature cases, the lyotropic liquid crystalline nanoparticles are able to demonstrate superior in vitro and in vivo performance, enhancing the therapeutic efficiency in a diverse range of applications, being administrated by various routes. The studied liquid crystalline nanosystems also exhibit great versatility and are strictly influenced by the environmental and formulation parameters, a characteristic that we can exploit towards the “on-demand” controlled drug delivery, as well as the development of stimuli responsive systems. As a conclusion, we should mention that there is an emerging need for continuous development of smart innovative biomaterials (surfactants, polymers, etc.) that will upgrade and further functionalize the already existing systems. Moreover, there is a great challenge regarding the careful and rational design that should be carried out during the development process of such systems by taking into account all the different interactions that may take place among the constituting biomaterials or the effects of external parameters, such as the environment. Finally, the application of robust and standardized characterization techniques is absolutely necessary in order to precisely evaluate the characteristics of such complex structures.

## Figures and Tables

**Figure 1 pharmaceuticals-15-00429-f001:**
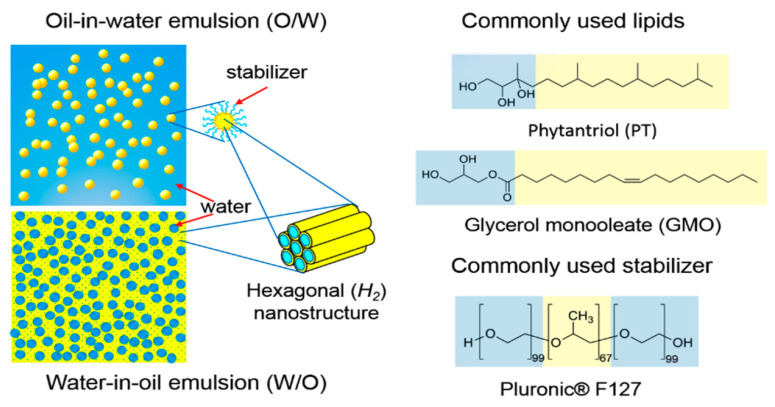
**Left panel**: Kinetically stabilized oil-in-water (O/W) and water-in-oil (W/O) nanostructured emulsions comprising a self-assembled lipid nanostructure. Apart from the illustrated hexagonal *H_2_* nanostructure, other types of nanostructures, including bicontinuous cubic *Pn3m* or *Im3m*, micellar cubic *Fd3m*, hexagonal (*H_2_*), or inverse micelles (*L_2_*), can be formed in these emulsions. **Right panel**: Chemical structures of the commonly used lipids and stabilizer. Adapted from Kulkarni [3].

**Figure 2 pharmaceuticals-15-00429-f002:**
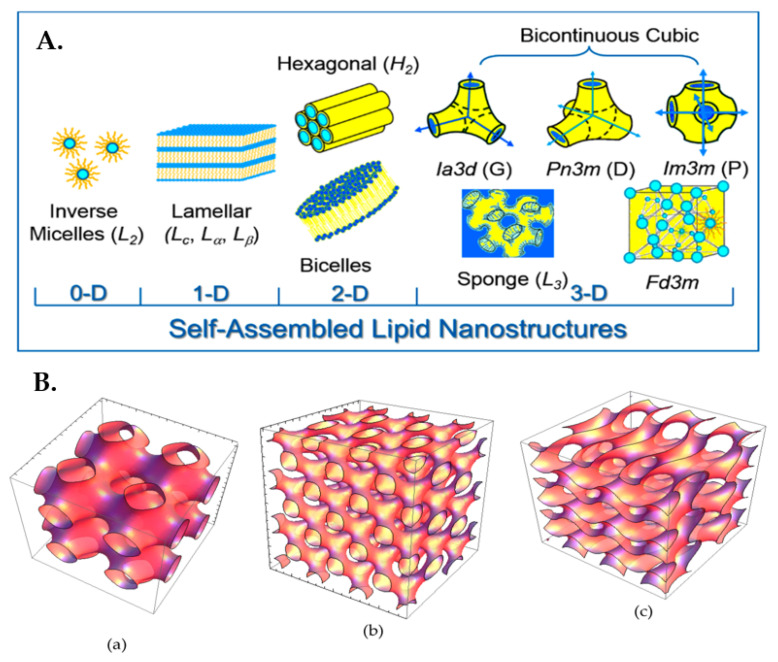
(**A**) Thermodynamically stable self-assembled lipid nanostructures. Adapted from Kulkarni [3]. (**B**) Three-dimensional organizations of cubic liquid crystalline phases: (**a**) primitive cubic (*Im3m*/Q^IIP^), (**b**) bicontinuous double diamond cubic (*Pn3m*/Q^IID^), and (**c**) bicontinuous gyroid cubic (*Ia3d/*Q^IIG^) types. Adapted from Rakotoarisoa et al. [13].

**Figure 3 pharmaceuticals-15-00429-f003:**
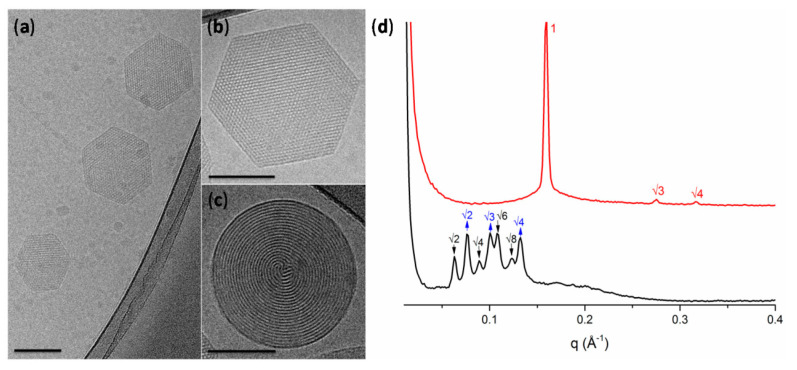
(**a**–**c**) Cryo-TEM images of hexosomes (all scale bars represent 100 nm); (**d**) SAXS profiles of the lipid-based liquid crystals without (black) and with undecylenic acid (red). Black, down arrows show the reflections of the *Im3m* cubic phase; blue, up arrows show the reflections of the *Pn3m* cubic phase. The reflections are annotated above the Bragg peaks. Adapted by Mionić Ebersold et al. [51].

**Figure 4 pharmaceuticals-15-00429-f004:**
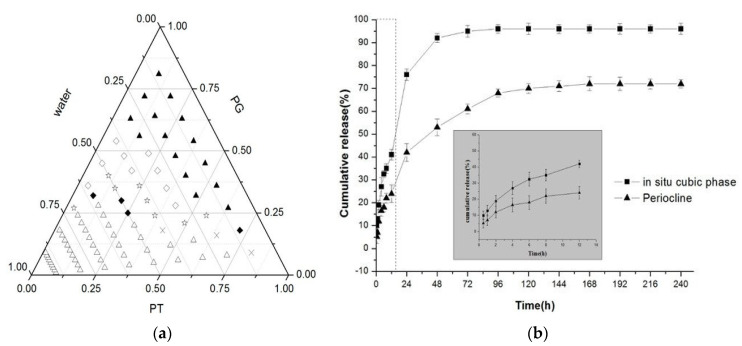
(**a**) Phase behavior of the PHYT-PG-water system. ▲ Isotropic solution; ♢ emulsion; ☆ emulsion + lamellar phase; ♦ lamellar phase; × lamellar + cubic phase; ∆ cubic phase. (**b**) Drug release studies of minocycline hydrochloride-loaded in situ cubic liquid crystal and Periocline^®^. Adapted by Yang et al. [128].

**Figure 5 pharmaceuticals-15-00429-f005:**
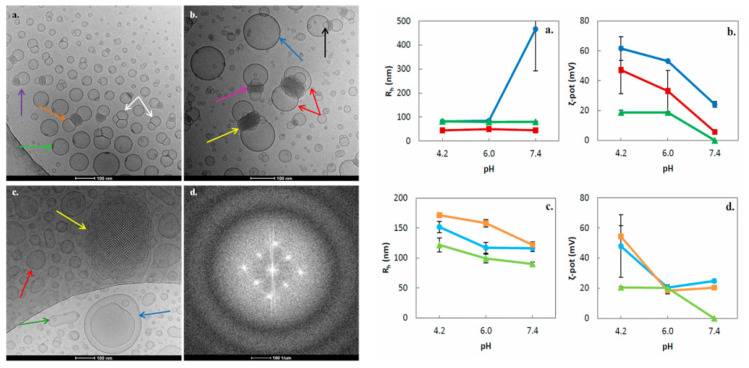
(Left Panel, **a**–**d**) Cryo-TEM images of liquid crystalline nanosystems with PDMAEMA-b-PLMA. (Right Panel, **a**–**d**) Physicochemical characteristics of the nanosystems depending on the pH of the dilution medium. Adapted by Chountoulesi et al. [38].

**Table 1 pharmaceuticals-15-00429-t001:** Summary of the reported case studies, regarding the applications of non-lamellar LLCN as drug delivery nanosystems.

Formulation	Therapeutic Agent	Therapy	Route of Administration	Notes	Reference
GMO:F127	Doxorubicin	Anticancer (glioblastoma)	In vitro	pH-dependent drug release	Nazaruk et al. [54]
Phosphatidyl choline:glycerol dioleate:Tween 80	Docetaxel	Anticancer (prostate cancer)	Intravenous	Better tumor regression compared to commercial	Cervin et al. [77]
GMO:F127:mPEG_2k_DSPE	Paclitaxel	Anticancer	Intravenous	PEGylation enhances the safety and efficacy of GMO systems	Jain et al. [78]
GMO:F127	5-Fluorouracil	Anticancer	Subcutaneous	Enhanced biodistribution	Nasr et al. [81]
MO:F127:DSPE-PEG-maleimide:EGFR antibodies	Paclitaxel	Anticancer (ovarian cancer)	Intraperitoneal	Enhanced cancer cytotoxicity	Zhai et al. [79]
GMO:F108	Camptothecin	Anticancer	In vitro	Increased targeting	Caltagirone et al. [82]
MO:F127	Cisplatin, Paclitacel, Dual	Anticancer	In vitro	Sustained drug release	Zhang et al. [83]
PHYT:DOTAP:F127	5-Fluorouracil	Anticancer	In vitro	Enhanced cytotoxicity in breast cancer cells	Astolfi et al. [84]
5-FCOle:F127:ethanol	5-Fluorouracil	Anticancer	Via orogastric gavage	Self-assembled amphiphile prodrugs	Gong et al. [85]
5-FCPal/5-FCOle/5-FCPhy:F127	5-fluorouracil	Anticancer	Via orogastric gavage	Self-assembled amphiphile prodrugs	Sangella et al. [86]
Soy phosphatidylcholine: glycerol dioleate:Tween 80	Paclitaxel	Anticancer	Oral	Enhanced oral bioavailability than commercial	Zeng et al. [80]
GMO:F127 into gelling system (F127, F68, HPMC K4M)	Docetaxel	Anticancer	-	Thermoresponsive depot system	Rarokar et al. [97]
Odorranalectin-decorated-GMO-F127	Gly14-humanin (S14G-HN) peptide	Alzheimer’s therapy	Intranasal (to brain)	Enhanced therapeutic effects	Wu et al. [123]
GMO:F127:Tween 80 in gellan gum or polyox gel	Risperidone	Schizophrenia	Intranasal (to brain)	Enhanced bioavailability and permeation	Abdelrahman et al. [121]
GMO:F127:Tween 80 or Cremophor RH 40	Piperine	Alzheimer’s therapy	Intranasal (to brain)	Sustained drug release	Elanggar et al. [122]
PHYT:F127	Amphotericin B	Antifungal	Oral	Enhanced oral bioavailability	Yang et al. [87]
PHYT:F68	Doxorubicin-CoQ10	Anticancer	Oral	Preventing cardiotoxicity	Swarnakar et al. [95]
GMO:F127:ethanol:Propylene glycol	Insulin	Diabetes	Oral	Taken up by Caco-2 cells	Chung et al. [89]
GMO:F127	Cyclosporine A	Antibiotics	Oral	Enhanced oral bioavailability compared to commercial	Lai et al. [90]
GMO:F127	Amphotericin B	Antifungal	Oral	Enhanced permeation in Caco-2 cells, enhanced oral bioavailability	Yang et al. [93]
GMO:F127:sorbitol	Tamoxifen	Anticancer	Oral	Enhanced oral bioavailability	Nasr and Dawoud [91]
PHYT:F127	Cinarizine	Model drug	Oral	Sustained drug release	Nguyen et al. [88]
GMO:F127	Spironolactone, nifedipine	Antihypertensive	Oral	Enhanced oral bioavailability	Ali et al. [92]
PHYT:F127:propylene glycol	Cefpodoxime proxetil	Antibiotic	Oral	Taste-making for pediatric patients	Fan et al. [96]
GMO:dextran:Eudragit^®^ L100–55 microcontainers	OVA and Quil-A	Vaccine	Oral	Improve the humoral response to oral boosters	von Halling Laier et al. [94]
GMO:F127:glycerol	Dexamethasone	Anti-inflammatory	Ocular	Enhanced bioavailability and unaffected corneal structure	Gan et al. [112]
GMO:F127:glycerol	Flurbiprofen	Anti-inflammatory	Ocular	Enhanced bioavailability compared with eye drops	Han et al. [113]
GMO:F127	Pilocarpin nitrate	Glaucoma	Ocular	Prolonged effect compared with commercial and controlled delivery	Li et al. [114]
GMO:F127	Brinzolamide	Glaucoma	Ocular	Better ocular bioavailability, and patient compliance compared to commercial	Wu et al. [115]
GMO:F127:glycerine	Timolol	Glaucoma	Ocular	Enhanced corneal permeability and bioavailability compared to commercial	Huang et al. [118]
GMO:F127	Cyclosporine A	Antibiotics	Ocular	Excellent ocular tolerance	Chen et al. [116]
GMO:F127	Tetrandrine	Glaucoma	Ocular	Enhanced ocular bioavailability	Liu et al. [117]
PHYT:F127 in thermo-gelling chitosan solution	Ciprofloxacin	Antimicrobial	Ocular	Improved eye permeation, prolonged ocular retention time, and enhanced antimicrobial activity compared to commercial	Alharbi et al. [119]
GMO:oleic acid: polyethylenimine (PEI)/oleylamine (OAM)	siRNA	Various	Topical skin	Without skin irritation	de Carvalho Vicentini et al. [109]
GMO:Tween 20	Celecoxib	Anti-inflammatory	In vitro skin	Enhanced skin permeation	Estracanholli et al. [110]
MO:F127	Cyclosporin A	Antibiotic	Topical skin	Without skin irritation	Lopes et al. [108]
GMO:F127:polyvinyl alcohol in chitosan/carbopol 940 hydrogels (cubogels)	Silver sulfadiazine	Burn therapy	Topical skin	Least side effects and better compliance than commercial	Morsi et al. [99]
GMO:F127 in carbopol 940/aloe vera hydrogels (cubogels)	Silver sulfadiazine and aloe vera	Burn therapy	Topical skin	Better bio adhesion and superior burn healing than commercial	Thakkar et al. [100]
GMO/PHYT:propylene glycol	δ-Aminolevulinic acid	Photodynamic therapy	Topical skin	Enhanced drug penetration	Bender et al. [102]
GMO:ethanol	Mitoxantrone	Melanoma	Topical skin	Non-invasion and no severe side effects	Yu et al. [103]
GMO:phospholipids:propylene glycol	Chlorin e6 or meso-Tetraphenylporphine-Mn(III) chloride	Photodynamic therapy (PDT) of melanoma	Topical skin	Biocompatible polymer-free cubosomes for potential application in both PDT and bioimaging	Bazylińska et al. [104]
GMO:F127:ethanol	Oregonin and Hirsutanonol	Atopic dermatitis	Topical skin	Enhanced skin permeation	Im et al. [106]
GMO:F127	Herbal extracts	Hair loss	Topical skin	Enhanced skin permeation	Seo et al. [107]
GMO:F127	Etodolac	Rheumatoid arthritis	Topical skin	Enhanced bioavailability	Salah et al. [105]
GMO:F127	antimicrobial peptide LL-37	Bacterial infections	Topical skin	Enhanced bactericidal effect without irritation	Boge et al. [101]
GMO:lactose	Salbutamol sulfate	Dry powder inhaler (DPI) formulation	Pulmonary	aerosolisation parameters as in commercial	Patil et al. [124]
Sorbitan monooleate:Tween 20:tocopherol acetate: phosphatidylcholine	Leuprolide acetate	In situ gelling system	Injectable	Sustained drug release (prostate cancer, endometriosis, and central precocious puberty)	Báez-Santos et al. [127] Ki et al. [126]
Sorbitan monooleate:Tween 20:tocopherol acetate: phosphatidylcholine	Entecavir	In situ gelling system	Injectable	Sustained drug release (hepatitis B)	Kim et al. [125]
PHYT:ethanol:vit E	Sinomenine hydrochloride	In situ gelling system	Intra-articular	Sustained drug release (rheumatoid arthritis)	Chen et al. [50]
PHYT:propylene glycol	Minocycline hydrochloride	In situ gelling system	Intra-periodontal pocket	Sustained drug release (periodontitis)	Yang et al. [128]
Stabilizer-free GMO	Antimicrobial peptide LL-37	Bacterial infections	In vitro	Antimicrobial, cytocompatible	Zabara et al. [22]
Monoglycerides (dimodan):F127	Undecylenic acid	Fungal infections (*Candida albicans*)	In vitro	Inhibition of fungal growth and filamentation, non-toxic in human cells	Mionić Ebersold et al. [51]
MO:F127	Indomethacin	Anti-inflammatory	Percutaneous	Depot effect on the epidermis	Esposito et al. [57]

**Table 2 pharmaceuticals-15-00429-t002:** Summary of the reported case studies with regards to the influence of environmental and formulation parameters in the internal nanostructures of non-lamellar LLCN, including the stimuli-responsive ones.

Lipidic Host Matrice	Additive	Phase Transition	Triggering Parameter	Reference
MO	linoleic acid	*Im3m* ↔ *H_II_* (*)	pH	Negrini and Mezzenga [41]
MO:F127	poly(L-lysine-iso-phthalamide) grafted with L-phenylalanine	Disruption of *Im3m*	pH	Kluzek et al. [42]
Monolinolein (MLO)	pyridinylmethyl linoleate	*Pn3m ↔ H_II_* (*)	pH	Negrini et al. [146]
MO	hydrophobically modified alginate (HmAL) and hydrophobically modified silk fibroin (HmSF)	Cubic with coacervate in water channels (*)	pH	Kwon and Kim [144]
GMO:F127	2-hydroxyoleic acid (2OHOA)	*Pn3m*, *H_II_* ↔ *Pn3m*, *Im3m* ↔ Lamellar	pH	Prajapati et al. [147]
GMO:oleic acid:F127	*Brucea javanica* oil, doxorubicin	*H_II_* → *Pn3m*, *Im3m* → Microemulsion (*)	pH	Li et al. [148]
PHYT:F127	decyl betainate chloride cleavable surfactant	lamellar-to-*Im3m*-to-*H_II_* (*)	pH	Ribeiro et al. [149]
1,2-Dimyristoyl-*sn*-glycero-3-phosphocholine (DMPC)	poly(acrylic acid)-dimyristoyl-*sn*-glycero-3-phosphoethanolamine (PAA-DMPE)	Swollen lamellar to cubic	pH, ionic strength	Crisci et al. [145]
GMO/PHYT:F127	poly(2-(dimethylamino)ethyl methacrylate)-b-poly(lauryl methacrylate) (PDMAEMA-b-PLMA)	Structure dependant on formulation, pH-responsive ζ-potential, and fractal dimension	pH, temperature	Chountoulesi et al. [30,38]
MO	fatty acid (oleic acid/vaccenic acid/gondoic acid/erucic acid/nervonic acid)	*Fd3m* → *H_II_*	pH, ionic strength, Temperature	Fong et al. [37]
GMO:diglycerol monoleate (DGMO)	poly(N-isopropylacrylamide) (pNIPAM) nanoparticles	*Pn3m*	Temperature	Dabkowska et al. [156]
Monolinolein (MLO):F127	-	*Pn3m ↔ H_II_ ↔ L_2_*	Temperature	de Campo et al. [151]
PHYT	laponite	*Pn3m ↔ L_2_*	Temperature	Muller et al. [152]
GMO/PHYT	vitamin E acetate	*Pn3m* ↔ *H_II_* (*)	Temperature	Fong et al. [63]
MO:cholesterol	dioleoyl-phosphatidylserine (DOPS) or dioleoyl-phosphatidylglycerol (DOPG)	Highly swollen *Im3m* (varying lattice parameter)	Temperature and pressure	Barriga et al. [155]
MO	dioleoyl-phosphatidylglycerol (DOPG)	*L_α_* → cubic	Ionic strength (Ca^2+^ cations)	Awad et al. [40]
PHYT:F127	sodium bis(2-ethylhexyl)sulfosuccinate (AOT), didodecyldimethylammonium bromide (DDAB)	Lamellar ↔ *Im3m* ↔ *Pn3m*	Ionic surfactant content, ionic strength	Liu et al. [39]
PHYT:F127	didodecyldimethylammonium bromide (DDAB)	Liposomes → cubosomes	Ionic strength	Muir et al. [67]
MO	dioleoyl-phosphatidylglycerol (DOPG)	*L_α_* → *H_II_*	Ionic strength (Ca^2+^ cations)	Yaghmur et al. [66]
PHYT:F127:5-fluorouracil	didodecyldimethylammonium bromide (DDABr)	*Pn3m ↔ Im3m ↔* Lamellar	Ionic content	Astolfi et al. [69]
Monolinolein (MLO)	outer membrane protein F (OmpF)	Topological interconnectivities between the aqueous nanochannels (*)	Protein incorporation	Zabara et al. [70]
MO	distearoylphosphatidylglycerol (DSPG)	Swollen cubic phases	Charged lipid	Engblom et al. [71]
MO	sucrose stearate	*Pn3m → Im3m*	Hydration-enhancing effect	Negrini and Mezzenga [73]
Monoacylglycerols and phospholipids	-	Ultra-swollen bicontinuous cubic phases of *Ia3d*, *Pn3m* and *Im3m*	Crystallize proteins with small extracellular domains (ECDs)	Zabara et al. [76]
MO:cholesterol	1,2-dioleoyl-sn-glycero-3-phospho-l-serine (sodium salt) (DOPS), 1,2-dioleoyl-sn-glycero-3-phospho-(1′-rac-glycerol) (sodium salt) (DOPG), 1,2-dioleoyl-sn-glycero-3-phosphate (sodium salt) (DOPA) charged phospholipids	Swollen cubosomes	Electrostatic tuning	Barriga et al. [74]
MO:cholesterol	phospholipids with PC, PE, and PS headgroups and saturated chain lengths from C12 to C18 [lauryl (C12); myristyl (C14); palmityl (C16); stearyl (C18)] except for the singly unsaturated (C18:1) oleoyl chain	Swollen cubosomes	Increase in curvature of the MO bilayer	Sarkar et al. [75]
GMO:vit E	d-α-tocopheryl poly(ethylene glycol)2000 succinate (TPGS-PEG2000) and thymoquinone	Coexisting *Fd3m* and *H_II_*-to-*Fd3m* or inverse micellar (*L_2_*)	Lipids ratio, presence, and concentration of stabilizer and drug	Yaghmur et al. [49]
PHYT:F108	1,2-distearoyl-*sn*-glycero-3-phosphoethanolamine-n-[amino(polyethylene glycol)-2000] (DSPE-PEG2000) and vitamin E acetate	Cubosomes → *H_II_* hexosomes → time-dependent growth of swollen hexagonal phase	Human monocytic cells (THP-1)	Tan et al. [45]
MO:TPEG_1000_ amphiphile:fish oil:curcumin	Catalase enzyme	*Pn3m* and *Im3m→ Im3m*	Enzyme presence	Rakotoarisoa et al. [13]
PHYT:F127	-	*Pn3m* with *H_II_* → neat *H_II_*	Human plasma	Azmi et al. [43]
Soy phosphatidylcholine: citrem	-	*Pn3m*, *H_II_ →* swollen ones	Human plasma	Azmi et al. [44]
GMO/PHYT:F127	Fetal bovine serum	Acute size reduction	Serum proteins	Chountoulesi et al. [9]
PHYT	Vitamin E acetate	Suppresses the temperature of *Q_II_*-to-*H_II_*-to-*L_2_* transitions	Vitamin E acetate	Dong et al. [48]

(*) Controlled drug release was experimentally testified and confirmed.

## Data Availability

Data sharing not applicable.

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
