# Peer review of "Lyotropic Liquid Crystalline Nanostructures as Drug Delivery Systems and Vaccine Platforms"

_pharmaceuticals, 2022, doi:10.3390/ph15040429_

Round 1

Reviewer 1 Report

The title of the manuscript (MS) gives an impression to readers that authors will give a general view of the subject, but reading the content of the MS, this impression seems to me not fully satisfied. Authors insist in the concept of "non-lamellar lyotropic liquid crystalline nanosystems". There are many different structures of liquid crystals that are "non-lamellar", e.g., nematics (uniaxial and biaxial), hexagonal, cubic, sponge phase. Why insist in the concept of non-lamellar instead to be assertive and give the characteristics of the structure of the mesophase under discussion? Lipid emulsions are used as drug target for many years by the group of R.C. Maranhao, and are not discussed in the MS. Concerning the techniques employed nowadays, the mention of SAXS is poorly made. New developments on the SAXS analysis concerning, e.g., the inverse Fourier transformation allows the description of all the scattering curve, not only the positions and widths of the Bragg peaks, furnishing a complete description of the structure. This type of approach is not discussed in the MS (see, e.g., C.L.P. Oliveira works). In summary, I see potential in he MS, however, major revisions are needed.

Author Response

We agree that liquid crystals are a wide category of the state of matter between liquids and crystals. However, in order to restrict our research within the purposes of drug delivery nanosystems, we chose to describe categories of liquid crystals that exhibit intense pharmaceutical interest as drug delivery nanoplatforms and vaccines. For example, nematics are thermotropic liquid crystals and not lyotropic and most significantly they are not applied as drug delivery nanosystems. In contrast, cubic and hexagonal phases are the most commonly used non-lamellar lyotropic liquid crystalline mesophases. Moreover, non-lamellar structures, as cubic and hexagonal phases, were chosen, because they exhibit high grade of internal organization and thus they exhibit significant pharmaceutical advantages that are analyzed in detail in the manuscript. Regarding the characteristics of the structure of the mesophase under discussion, we analytically described it in the introductive paragraph 2, along with its formation process, accompanied by the respective illustrative images and subsequently we focused on their pharmaceutical applications in the following paragraphs. Regarding the lipid emulsions, although the lipid emulsions are used as drug target for many years, they exhibit lamellar structure of lower organization of one simple lipid bilayer and that is why we chose to exclude them from the present review study. Concerning the SAXS technique, we discussed various literature case studies, where SAXS was applied to cubic and hexagonal liquid crystals, but we did not focus on SAXS technical details, in order not to diverge from our main pharmaceutical scope of the review paper. For example, to the best of our knowledge, we did not find any reference on cubic and hexagonal liquid crystals, where inverse Fourier transformation is applied.     

Reviewer 2 Report

The review by Chountoulesi et al. is very accurate and complete. I really appreciated the effort to present both the characterization work and the applications, also considering very recent applicative works.
I would have liked to have found citations of now classic but always current works by Vittorio Luzzati's group (e.g., JMB, 1988, 204, 165-189 and JMB 1993, 229, 540-551). 

Author Response

Thank you very much for your encouraging comments. The proposed references from Vittorio Luzzati's group were also cited in the revised form of the manuscript.

Reviewer 3 Report

The review article proposed by Chountoulesi et al., gives a complete overview on lyotropic liquid crystalline nanostructures and their use as drug delivery systems.

The manuscript is comprehensive, well-structured, and clear. An excellent revue!

Author Response

Thank you very much for your highly positive comments.

Round 2

Reviewer 1 Report

I still have some problems with author's text and response presented.

1) In their response is written: "...nematics are thermotropic liquid crystals and not lyotropic...". This is wrong! Lyotropic nematics (uniaxial and biaxial) exist, being firstly reported (a uniaxial) in 1967. After that, many studies on lyotropic nematics were performed. There are books about the Physics of lyotropics describing in details the lyotropic polymorphism.

2) in their response is written: "...although the lipid emulsions are used as drug target for many years, they exhibit lamellar structure...". This is also wrong! Low-Density Emulsions (LDE) are globular, not lamellar as claimed by authors, and are used to carry drugs.

3) the proposed concept of "non-lamelar" lyotropics seems to me misleading and should be removed from the text. Instead of use a negative term, it would be better to say exactly which mesophase authors are speaking about. 

In summary, I still recommend revisions of the text.

Round 3

Reviewer 1 Report

Author's responses seems to me adequate. I still keep my concern abou the claimed category of "non-lamellar liquid crystals". May be a jargon of this particular field of research, but for me is inadequate. Just an example: imagine that I would like to describe some behaviours of "yellow frogs". If I say in the abstract of my paper that I will discuss the behaviour of "non-blue frogs", this information is not enough for readers. But OK, I see that authors are happy with this characterisation, so, I recommend the publication of the ms as it is now.